# Investigating the Effect of Surface Hydrophilicity on the Destiny of PLGA-Poloxamer Nanoparticles in an In Vivo Animal Model

**DOI:** 10.3390/ijms241914523

**Published:** 2023-09-25

**Authors:** Teresa Silvestri, Lucia Grumetto, Ilaria Neri, Maria De Falco, Sossio Fabio Graziano, Sara Damiano, Daniela Giaquinto, Lucianna Maruccio, Paolo de Girolamo, Fabrizio Villapiano, Roberto Ciarcia, Laura Mayol, Marco Biondi

**Affiliations:** 1Department of Pharmacy-Pharmaceutical Sciences, University of Bari Aldo Moro, Via E. Orabona 4, 70125 Bari, Italy; teresa.silvestri@uniba.it; 2Department of Pharmacy, University of Naples Federico II, Via D. Montesano 49, 80131 Naples, Italy; lucia.grumetto@unina.it (L.G.); ilaria.neri@unina.it (I.N.); mabiondi@unina.it (M.B.); 3National Institute of Biostructures and Biosystems (INBB), Viale delle Medaglie d’Oro 305, 00136 Rome, Italy; 4Department of Biology, University Federico II of Naples, Via Cinthia 26, 80125 Naples, Italy; madefalco@unina.it; 5Department of Veterinary Medicine and Animal Productions, University of Naples Federico II, Via Federico Delpino 1, 80137 Naples, Italydaniela.giaquinto@unina.it (D.G.); lucianna.maruccio@unina.it (L.M.);; 6Department of Advanced Biomedical Sciences, University of Naples Federico II, Via D. Montesano 49, 80131 Naples, Italy; 7Interdisciplinary Research Centre on Biomaterials (CRIB), Piazzale Tecchio 80, 80125 Naples, Italy

**Keywords:** nanoparticles (NPs), in vivo, biodistribution, poly(lactic-co-glycolic) acid (PLGA), poloxamers, polymeric blend

## Abstract

This study aimed to examine the impact of different surface properties of poly(lactic-co-glycolic) acid (PLGA) nanoparticles (P NPs) and PLGA-Poloxamer nanoparticles (PP NPs) on their in vivo biodistribution. For this purpose, NPs were formulated via nanoprecipitation and loaded with diphenylhexatriene (DPH), a fluorescent dye. The obtained NPs underwent comprehensive characterization, encompassing their morphology, technological attributes, DPH release rate, and thermodynamic properties. The produced NPs were then administered to wild-type mice via intraperitoneal injection, and, at scheduled time intervals, the animals were euthanized. Blood samples, as well as the liver, lungs, and kidneys, were extracted for histological examination and biodistribution analysis. The findings of this investigation revealed that the presence of poloxamers led to smaller NP sizes and induced partial crystallinity in the NPs. The biodistribution and histological results from in vivo experiments evidenced that both, P and PP NPs, exhibited comparable concentrations in the bloodstream, while P NPs could not be detected in the other organs examined. Conversely, PP NPs were primarily sequestered by the lungs and, to a lesser extent, by the kidneys. Future research endeavors will focus on investigating the behavior of drug-loaded NPs in pathological animal models.

## 1. Introduction

The advent of nanoparticle-based systems has prompted a significant paradigm shift in cancer treatment and drug delivery [1]. Still to date, poly(lactide-co-glycolic acid) (PLGA) has garnered a significant deal of attention as a material of choice for the production of nanoparticles (NPs). Indeed, as highlighted in a recent review, among the various biodegradable synthetic polymers such as polyanhydrides, polyorthoesters, polyphosphazenes, polyamidoesters (heterochain polymers) and polycyanoacrylates (homochain polymers), thermoplastic aliphatic polyesters have always dominated in the field of controlled drug release, due to their wide range of favorable physicochemical and biological properties [2]. More in detail, PLGA has been approved by the US FDA for various therapeutic applications due to its biocompatibility, complete biodegradability, and tunable properties [2,3]. Furthermore, an important property of block copolymers is that, by changing the content of the two monomers in the copolymer, it is possible to modulate their physicochemical properties. For example, the biodegradation rate of PLGA can be suitably tailored according to the application purposes, with the in vivo resorption period of PLA (100% LA) ranging between 12 and 24 months, while that of PGA (100% GA) ranging only 6–12 months. This variation in biodegradation rate is due to the poorer hydrophilicity and slower degradation rate of PLA and the relatively higher hydrophilicity and faster degradation rate of PGA. With an LA:GA ratio of 50:50, PLGA copolymers have a high rate of degradation that slows down as the proportion of LA increases from 50 to 100 and that of GA decreases from 50 to zero [2].

However, the clinical translation of PLGA-based NPs necessitates a thorough understanding of their behavior in biological systems, including their biodistribution, clearance, and potential accumulation in off-target organs [4].

To overcome limitations associated with the rapid clearance and non-specific uptake of NPs by the reticuloendothelial system (RES), surface modification strategies have been employed [5]. Among these, the surface coating of NPs with hydrophilic moieties has emerged as an effective means to discourage their rapid phagocytosis in vivo, therefore enhancing their stealth properties and prolonging their circulation time [6].

In this context, the incorporation of a hydrophilic polymer such as polyethylene glycol (PEG) on NP’s surface is expected to hinder the adsorption of opsonins, hence enhancing NP stability in physiological conditions. This fosters the extended circulation of NPs, thereby increasing their passive accumulation to tumors [7].

To this aim, we have recently formulated NPs made up of an equiponderal mixture of PLGA and poloxamers. Poloxamers, also known by the brand name Pluronic, are triblock copolymers made up of poly(ethylene oxide)–poly(propylene oxide)–poly(ethylene oxide) (PEO–PPO–PEO), displaying amphiphilic properties taking advantage of the presence of hydrophilic EO and hydrophobic PO segments on a polymer backbone. There are many types of poloxamers on the market that differ in the number of units of EO and PO; however, in previous studies we have adjusted the NP formulation, in terms of both the type of poloxamers used and the relative ratio of the various polymers. This adjustment was made in order to obtain optimized NPs in terms of size, polydispersity and zeta potential [8,9]. The production of NPs was based on the formation of an amphiphilic polymeric blend of PLGA and poloxamers by means of emulsion techniques followed by solvent evaporation, all with an external aqueous phase. Under these experimental conditions, the hydrophilic segments of the poloxamers spontaneously orient toward the aqueous phase, therefore allowing an increase in NP surface hydrophilicity. In two recent publications, we showed in an in vitro cellular experiment on mesothelioma cells, that NPs were not toxic and curcumin encapsulated in such NPs was released for days. This subsequently allowed the amplification of a block of mesothelioma cancer cells in the G_0_/G_1_ phase of the cell cycle for up to 72 h [8,9].

Despite the lasting interest in PLGA-based NPs, there remains a critical knowledge gap regarding their biodistribution and organ-specific targeting in healthy organisms [10]. Investigating the in vivo distribution of NPs in healthy mice is important for evaluating their safety, optimizing therapeutic dosing regimens, and minimizing potential off-target effects [11].

Hence, in this work we intended to validate our previous encouraging results also in vivo, by carrying out biodistribution tests of NPs in healthy mice. Specifically, the aim was to verify whether the presence of poloxamers in the formulation of PLGA NPs, leads to a different in vivo biodistribution. For this purpose, fluorescent NPs were formulated by loading diphenylhexatriene (DPH) as a dye to the organic phase of the primary emulsion used to formulate NPs, allowing for NP visualization in vivo. The obtained NPs underwent comprehensive characterization, encompassing their morphology, technological attributes, DPH release rate, and thermodynamic properties. The NPs were then administered by intraperitoneal injection, and, at scheduled time points, the animals were euthanized, and serum was withdrawn. Meanwhile, the liver, lungs, kidney, were removed, divided in aliquots and frozen at −20 °C or fixed in paraformaldehyde. Organs were processed for analysis via immunofluorescence studies. Histological analyses of the excised organs were carried out to qualitatively evaluate the in vivo biodistribution of both fluorescent bare PLGA NPs and those obtained by adding poloxamer(s) to the formulation.

## 2. Results and Discussion

This study focused on the effect of surface properties of poly(lactic-co-glycolic) acid (PLGA) -based nanoparticles (NPs) on their biodistribution in heathy mice. As described beforehand, the surface composition of the NPs is influenced by the presence of hydrophilic units in the organic phase of the primary emulsion [8,9]. In particular, in previous publications we have demonstrated that the addition of amphiphilic poloxamers to the organic phase of NPs results in the spontaneous arrangement of hydrophilic ethylene oxide (EO) units on NPs, and this is expected to discourage the adsorption of serum proteins [8,9]. Thus, in this work we aimed to assess whether the different surface properties of PLGA NPs (P NPs) and PLGA-Poloxamer nanoparticles (PP NPs) can affect their in vivo biodistribution in healthy mice. The use of a production technique in emulsion followed by nanoprecipitation allows an effective control of the size and size distribution of the nanocarriers [12]. Indeed, in this study NPs with a mean diameter below 200 nm were obtained in all cases (Table 1).

Specifically, PP NPs present a lower mean diameter than P NPs, and this is ascribable to the presence of amphiphilic poloxamers in the organic phase. This in turn is associated with a reduction in interfacial tension between the continuous and dispersed phases in the emulsion used to prepare the NPs. Therefore, this results in the formation of smaller droplets and, consequently, of NPs with smaller mean size. It must also be specified that PdI values were fairly low in both formulations. The presence of poloxamers in PP NPs also leads to a slight increase in ζ potential values due to a partial masking of PLGA carboxylic groups on the NP surface. We have previously investigated the superficial composition of PLGA-poloxamer microparticles MPs by means of electron dispersive X-ray spectroscopy (EDS) [13]. In more detail, EDS analyses were performed on the raw materials (poloxamers and PLGA), as well as on PLGA (P) and PLGA-Poloxamer (PP) MPs. The elemental composition of PLGA powder was found to closely resemble that of PP MPs. However, a slightly higher carbon percentage was found in the latter case, indicating that some poloxamers were present on the MP surface, even if PLGA prevails at the micro-scale. Moreover, in a previous work, we found that, at the nanoscale, the presence of poloxamers did actually enhance the superficial hydrophilicity of nanoparticles based on PLGA and poloxamers. This indicates that the superficial arrangement of hydrophilic segments of poloxamers is enhanced also at the nano-scale [8]. Results reported in Appendix A show that DPH entrapment efficiencies were >96% for both P and PP NPs.

SEM micrographs (Figure 1) show that the produced particles were uniformly spherical and well dispersed, with a smooth surface morphology.

As shown in Figure 2 and summarized in Table 2, the thermograms of blank and DPH-loaded P and PP NPs are reported. Both NP formulations show a glass transition around 49 °C followed by an endothermic hysteresis peak. Due to the amorphousness of PLGA, no endothermic melting peaks were observed. As can be noted regarding DPH-loaded NPs, the peak is shifted on the left compared to unloaded formulations. This can be correlated to a slight plasticizing effect of the dye. In addition, the thermograms of both PP NPs show a melting peak, ascribed to the fusion of the crystalline phase, around 51.6 °C and 47.8 °C, respectively. Again, the presence of DPH causes a plasticizing effect, thereby reducing the melting temperature.

To assess whether the detected fluorescence in vivo could be correlated with NP accumulation in the target organs, we performed in vitro DPH-release experiments in buffer solution within the same time frame as in vivo biodistribution experiments. Figure 3 shows the cumulative release of the DPH after four hours in the medium release.

The percentage of the released DPH is 13.2 ± 6.3% for P NPs and 0.06 ± 0.32% for PP NPs. The faster release displayed by P NPs is probably ascribable to the more hydrophobic nature of the PLGA polymeric matrix which enhances the free diffusion of the DPH through the porous architecture of NPs. Regardless, we can state that after four hours most of DPH remains within the NPs and therefore the fluorescent signal detected in the excised organs can be associated to NPs. It must also be underlined that DPH fluorescence is higher in a lipophilic milieu [14]. Consequently, it can be assumed that the quantified DPH in different organs (liver, kidney and lung summarized in Table 3, Table 4 and Table 5, respectively) is indicative of the NP amount.

As it can be seen from Table 6, NP accumulation in serum is very similar for both P and PP NPs. Contrariwise, only PP NPs are detectable in the analysed organs. These data were also corroborated by results of histological experiments shown in Figure 4. These outcomes indicated that PP NPs could be detected in the lungs (Figure 4a) and, to a lesser extent, within the kidneys (Figure 4b), indicating the prompt elimination of P NPs from these organs. As for PP NPs, the slight uptake in the lungs is clearly detectable in the intercellular spaces around pneumocytes composing alveolar structures and around blood vessels, whereas in the kidneys, PP NPs are located inside blood vessels and in the intercellular spaces around tubules. Figure 4c shows that in the liver, PP NPs could be detected only inside blood vessels.

These biodistribution outcomes can be related to the promoted surface hydrophilicity of PP NPs [15] and to the route of administration. Specifically, intraperitoneal administration of NPs is followed by rapid adsorption of plasma proteins after a brief retention within the peritoneal cavity [16]. The subsequent formation of a protein corona surrounding the NPs is expected to affect NP biodistribution. For example, opsonin adsorption is known to enhance NP recognition by the reticulo-endothelial system (RES) [17]. The rapid sequestration of NPs by RES can be delayed if non-ionic hydrophilic units such as the EO segments of poloxamers are present on the NPs, which confer steric stabilization to the nanodevices, as well as a partial reduction of surface charges.

According to the biodistribution data, it is possible to assume that both P and PP NPs are promptly sequestered and eliminated by Kupffer cells after administration (Figure 4c). Moreover, NP charge also slows their elimination from the body, which explains their persistence in the serum 4 h after administration.

The elimination of NPs through the kidneys is less likely, due to their relatively large size and negative zeta potential [18]. The localization of a small amount of PP NPs in the lungs and kidneys is not completely clear, but it is possible to hypothesize that alveolar and kidney macrophages may be involved. This is indicative of the fact that the biodistribution in healthy organs changes due to the different surface hydrophilicity of the formulated NPs. Overall, these findings encourage us to deepen the results found in this study by loading a hydrophobic drug in PP NPs targeted to the lungs, liver, or spleen to evaluate both the biodistribution in pathological organs and the effect of the loaded drug.

## 3. Materials and Methods

### 3.1. Materials

Equimolar poly(lactic-co-glycolic) acid (PLGA) (Resomer^®^ RG504H, poly(D,L-lactide-co-glycolide) (ester terminated; lactide:glycolide ratio 50:50; Mw = 38–54 kDa), KCl, NaCl, Na_2_HPO_4_, Ethanol (EtOH), 1,6-diphenil-1,3,5-hexatriene (DPH; CAS number 1720-32-7), Poloxamer 407 and 188 (also known as Pluronic F127 and Pluronic F68, respectively) were obtained from Sigma-Aldrich (Milan, Italy). MilliQ water was produced in-house (conductivity 0.055 μS∙cm^−1^ at 25 °C, resistivity equals 18.2 MΩ·cm). Acetone and acetonitrile, HPLC-analytical grade, were both from Sigma Aldrich (Milan, Italy). The wild-type mice of the CD1 strain supplier was Charles River Laboratories (Milan, Italy). 

### 3.2. Nanoparticle Preparation

Fluorescent nanoparticles (NPs) were fabricated by nanoprecipitation technique followed by organic solvent evaporation, as described in our previous works [8,9]. Briefly, NPs were produced without any chemical reaction, by merely exploiting a lipophilic gradient between an aqueous phase and an organic phase. In detail, a primary water-in-oil emulsion was obtained by sonicating (Misonix Sonicator 3000; 5 min, 4 W; Microtip 419; Misonix, NY, USA) 500 µL of an ethanol internal aqueous phase within an organic phase composed of a solution of either PLGA (namely P NPs) or PLGA:F68:F127 (1:0.5:0.5 weight ratio) in acetone (namely PP NPs). The overall polymer concentration was 3% *w/v* in all cases and 1 mg of the fluorescent dye 1,6-diphenyl-1,3,5-hexatriene (DPH). Thereafter, the obtained *W*_0_*/O* emulsion was poured through a syringe needle (22 G) by a syringe pump (Q = 333.3 μL/min; d = 11.99 mm) in 40 mL of an aqueous phase, containing F127 and F68 as surfactants (1:1 weight ratio; 0.0375 mg/mL each). Acetone was evaporated overnight by magnetic stirring (700 rpm) at room temperature and the resulting suspension was washed three times with double distilled water by centrifugation to withdraw the non-encapsulated DPH (10,000 rpm, 10 min; MKRO 300).

### 3.3. Nanoparticle Mean Diameter, Polydispersity Index and ζ Potential

Mean diameters and ζ potentials of NPs were determined through dynamic light scattering (DLS) analysis performed by Zetasizer Ultra (Malvern Instruments, Malvern, UK). For particle size measurements, NPs were suspended in ultrapure water. A total of five runs were performed for each experiment.

### 3.4. Morphological Analyses of Nanoparticles

Morphological investigation on P and PP NPs with or without DPH were performed using a Field Emission Scanning Electron Microscope (SEM) equipped with an Energy Dispersive Spectrometer (FESEM/EDS; Zeiss Merlin VP Compact coupled with Oxford Instruments Microanalysis Unit; Carl-Zeiss Strasse, Oberkochen, Germany). Samples were glued by a carbon tape film on a sample holder and then gold-metalized, using an automatic sputter coater (Agar Scientific ltd-Parsonage Lane, Stansed-Essex, UK), with an automatically controlled complete sequence of flush, leak, coat, and vent.

The INCA (Oberkochen, Germany) X-stream pulse processor and the INCA Energy software 5.05 were used to obtain datasets. Operative conditions were reported on each acquisition and were chosen to be consistent with the performed measurement.

### 3.5. Thermal Analyses

Analyses by differential scanning calorimeter (DSC) were carried out to investigate the interactions between the different polymers in NP formulations. Specifically, the tests were conducted on freeze-dried P and PP NPs, in the presence/absence of DPH using by DSC Q20 (TA Instruments, New Castel, DE, USA). The samples were loaded in aluminum pans sealed with Tzero lid and underwent a single scan, from 10 °C to 80 °C with a 5 °C/min heating rate. Measurements were performed under an inert nitrogen atmosphere, purged at a flowrate of 50.0 mL/min. 

### 3.6. Entrapment Efficiency of 1,6-diphenil-1,3,5-hexatriene (DPH)

The 1,6-diphenil-1,3,5-hexatriene (DPH) entrapment efficiency was determined by an indirect method by evaluating the difference between the initial DPH amount (*DPH_tot_*) and the unentrapped DPH in the supernatant (*DPH_sur_*), with respect to the total DPH mass used for NP preparation [19]. The following equation was used:(1)$$EE\left(\%\right)=\frac{{DPH}_{tot}-{DPH}_{sur}}{{DPH}_{tot}}$$

In detail, after organic solvent evaporation, NP suspensions were centrifuged three times (10,000 rpm, 10 min) and the recovered supernatant analysed via HPLC to detect the non-entrapped dye content after each centrifugation step. Stock standard solution of DPH (2 mg∙mL^−1^) was dissolved in acetone and used for dilutions required for calibration curve. Calibration curve was performed using dilution 25.0, 50.0, 100.0, 200.0, and 400 ng∙mL^−1^. All solutions were stored at 4 °C and kept in the dark until experiments.

HPLC analyses were performed using LC-20 AD apparatus (Shimadzu Corp., Kyoto, Japan) equipped with a fluorescence detector (model RF-20, Shimadzu Corp., Kyoto, Japan) set at the excitation wavelength (λEx) of 350 nm and at the emission wavelength (λEm) of 452 nm. The isocratic mobile phase utilized consists of acetonitrile:water MilliQ (85:15, *v/v*) and the flow rate was set to 0.8 mL∙min^−1^. All mobile phases were vacuum filtered through 0.45 μm nylon membranes (Millipore, Burlington, MA USA). The stainless-steel column was a Supelco, Ascentis^®^, C18, (250 × 4.6 mm, 5.0 µm particle size) with an Ascentis^®^ C18 Supelguard™ Guard Cartridge (20 × 4 mm, 5 μm particle size), and three times the loop volume, i.e., 60 μL, were injected at room temperature (i.e., 22 ± 2 °C). Data acquisition and integration were accomplished by Cromatoplus © 2011 software. The linearity of the response was assessed in DPH concentration ranging from 25.0 to 400.0 ng∙mL^−1^, with an R^2^ = 0.9949. Limit of detection (LOD) and limit of quantification (LOQ) of the methods resulted as 5.10 ng∙mL^−1^ and 25. 78 ng∙mL^−1^ respectively.

### 3.7. Release Kinetics of DPH in Phosphate Buffer Solution (PBS)

The release kinetics of DPH were determined in phosphate buffer solution (PBS), which emulates physiological conditions. Specifically, the buffer was prepared by dissolving 1.420 g of Na_2_HPO_4_, 0.201 g of KCl and 7 g of NaCl in 500 mL of double distilled water; after complete dissolution under magnetic stirring at room temperature, the pH was adjusted to 7.4 and the solution further diluted to 1 L final volume. The prepared buffer was filtered through a 0.45 µm membrane (RC). Afterwards, 200 µL of NP suspension were diluted in 1 mL of PBS and the samples were placed at 37 °C in a thermostatic bath under mild magnetic stirring (100 rpm). After 1 and 4 h, the suspensions were centrifuged (13,000 rpm, 15 min; MKRO 300) and the supernatant was analysed by HPLC for DPH quantification.

### 3.8. Animal Studies

The use and care of the animals in this work was approved by the Institutional Animal Care and Ethics Committee (Approval Number: 476/2018-PR) and carried out in accordance with the associated guidelines of the national law D.L. 26/2014 on the use of animals for research based on EU Directive 2010/63/EU.

Eight-week-old female wild-type mice of the CD1 strain were randomly distributed into two experimental groups (8 mice for each group) and housed in cages for at least 8 days under standard conditions (temperature 20 ± 2 °C and 12 h day/night cycles; relative humidity: 45–65%). The animals received a standard diet ad libitum.

At the time of experiment, 0.15 mL of fluorescent nanoparticles suspensions (0.15% *w*/*v*) were administered by intraperitoneal injection. At scheduled time points (1 and 4 h after injection), the mice were anesthetized with 2% isoflurane (Isotec 4, Palermo, Italy) and, after complete sedation, euthanized by cervical dislocation. 

After euthanasia, blood samples were withdrawn through the intracardiac route, collected in heparinized Vacu-test^®^ tubes and promptly centrifuged to separate the serum from other whole-blood components. Thereafter, the supernatant was transferred to a clean glass vial, frozen and immediately cryopreserved in refrigerator at −20 °C. For analytical tests, 475 μL of each sample were added to 1 mL of acetonitrile and maintained in ice at 4 °C for 5 min to allow proteins precipitation. Then, each sample was vortexed for 1 min and ultra-sounded for 30 s and finally centrifuged at 9500 rpm for 10 min. The supernatant was collected and injected into the HPLC; the results are the averages of at least three determinations.

In addition, liver, lungs, and kidneys were removed, weighed, and homogenized in isotonic KCl. An aliquot (0.490 g) of the homogenized solution was mixed with 2 mL of an acetone: formic acid (9:1 *v/v*) solution, and the resulting mixture was immediately vortexed for 10 min. The samples were then centrifuged at 6000 rpm for 10 min at 4 °C and the supernatant was collected and frozen at −20 °C. Before the analysis, samples were thawed and centrifuged at 12,000 rpm for 15 min; 20 μL of supernatants were collected and analysed by the HPLC system. Recovery extraction from sera and organs were assessed by spiking both, sera and organ homogenates, with known concentrations of the DPH, before the extraction procedure. 

### 3.9. Histological Analyses

To evaluate distribution of P and PP NPs, liver, lungs, and kidneys were fixed in Bouin for 24 h at room temperature, dehydrated in ascending alcohols, paraffin embedded and cut into 5 μm serial sections using a rotative microtome. Subsequently, the sections were subjected to routine histological analysis and stained with haematoxylin and eosin (H&E) [20]. Serial sections were observed with Axioshop epifluorescence microscope using a 40× objective and the acquisition software Axiovision 4.7 (Carl Zeiss, Milano, Italy).

### 3.10. Statistics

Results are reported as the mean of at least three replicates ± standard deviation (SD). One-way analysis of variance (ANOVA) and *t*-test were conducted for statistical analyses, and *p* values < 0.05 were considered statistically significant.

## 4. Conclusions

In conclusion, this study describes a comparison between the biodistributions of two nanoparticulate formulations after a single intraperitoneal injection. The results showed that both poly(lactic-co-glycolic) acid (PLGA) nanoparticles (P NPs) and PLGA-Poloxamer nanoparticles (PP NPs) can persist in serum in similar concentrations, but P NPs cannot be determined in the other organs under study. Conversely, PP NPs were sequestered by the lungs and, to a lesser extent, by the kidneys. To evaluate the potential of the formulations produced, it will be necessary to select a lipophilic-active molecule to be loaded into PP NPs. Future in vivo studies will concern the determination of the toxicity of the free drug and of the drug-loaded NPs for longer times, with a view to their preclinical validation as possible carriers for targeted chemotherapies.

The versatility of these nanoparticles (NPs) extends across various administration routes, opening non-invasive applications beyond cancer treatment. For instance, NPs can be integrated within hydrogels to breach the skin barrier in the treatment of psoriasis [21] and melanoma [22]. Additionally, NPs can be introduced through the nasal route via sprays or drops, effortlessly traversing the olfactory pathway to reach the brain and treat epilepsy [23]. Inhalation serves as another avenue for NPs, applicable in the management of asthma, chronic obstructive pulmonary disease and cystic fibrosis [24]. Altogether, the implications of this study encompass the design of NPs with finely tuned surface hydrophilicity, unlocking new dimensions in an array of targeted therapeutic approaches.

## Figures and Tables

**Figure 1 ijms-24-14523-f001:**
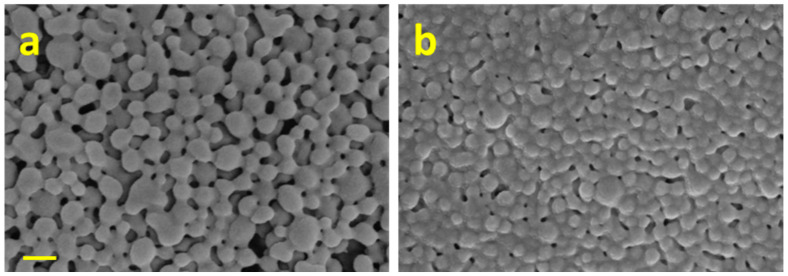
Scanning electron microscopy (SEM) images of: (**a**) poly(lactic-co-glycolic) acid (PLGA) nanoparticles (P NPs) and PLGA-Poloxamer nanoparticles (PP NPs); (**b**) PLGA-Poloxamer nanoparticles (PP NPs). The scale bar is 200 nm.

**Figure 2 ijms-24-14523-f002:**
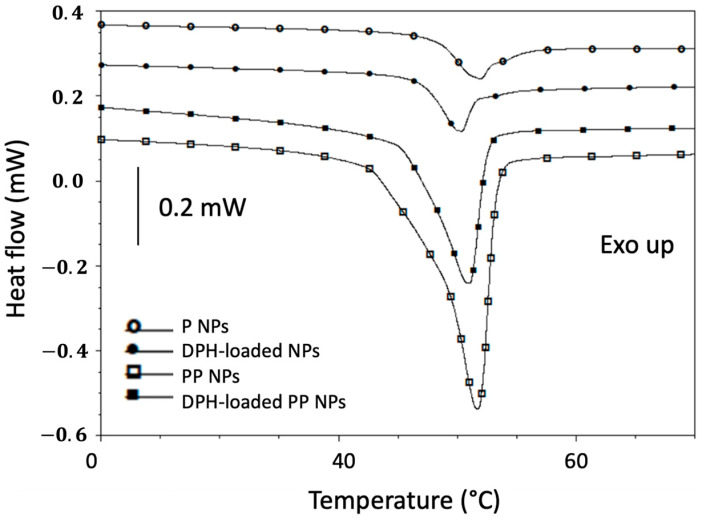
Thermograms of poly(lactic-co-glycolic) acid (PLGA) nanoparticles (P NPs) and PLGA-Poloxamer nanoparticles (PP NPs), with/without DPH. The exotherm is upwards.

**Figure 3 ijms-24-14523-f003:**
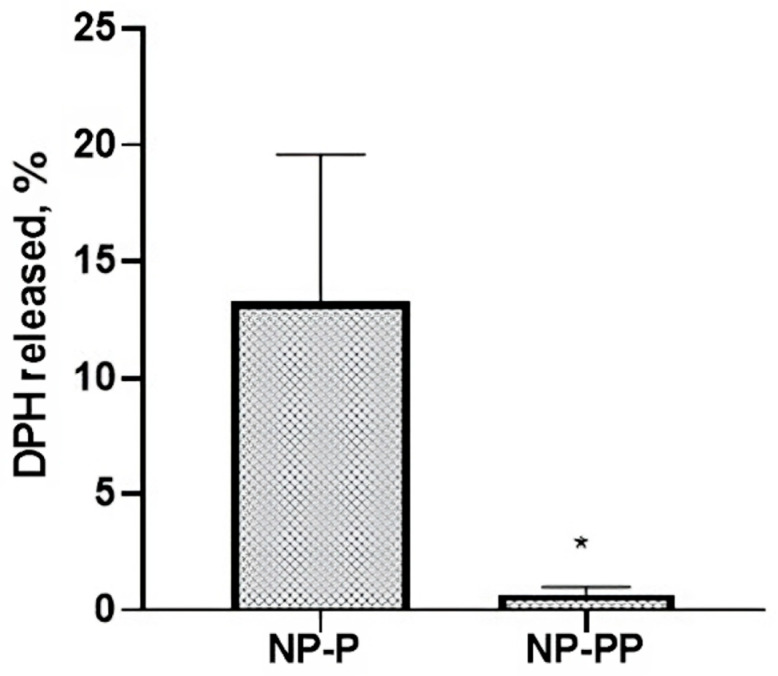
Percentage of DPH released from poly(lactic-co-glycolic) acid (PLGA) nanoparticles (P NPs) and PLGA-Poloxamer nanoparticles (PP NPs). Results are expressed as the mean ± SD of three replicates. Statistical analysis was performed by an unpaired *t*-test (* *p* < 0.05).

**Figure 4 ijms-24-14523-f004:**
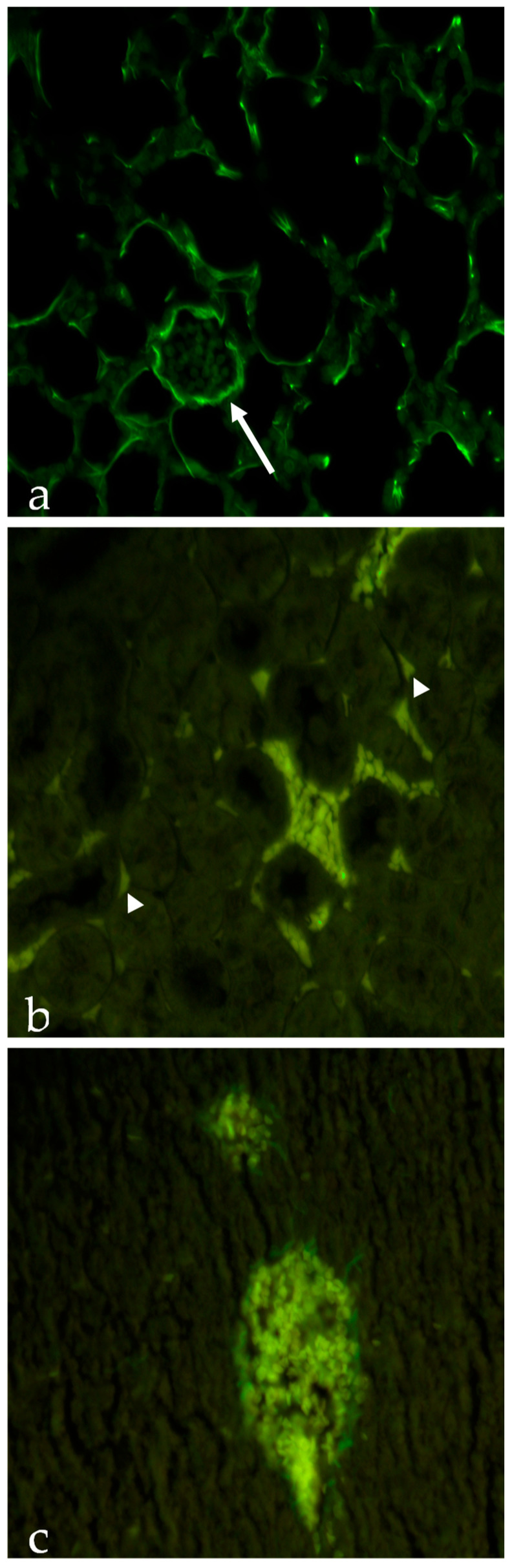
Photomicrograph of poly(lactic-co-glycolic) acid (PLGA)-Poloxamer nanoparticles (PP NPs): (**a**) in the lungs, PP NPs are clearly detectable in the intercellular spaces around pneumocytes composing alveolar structures and around blood vessels (arrow); (**b**) in the kidneys, PP NPs are located inside blood vessels and in the intercellular spaces around tubules (arrowheads); (**c**) in the liver, PP NPs can be detected only inside blood vessels. Original magnification was 40× in all micrographs.

**Table 1 ijms-24-14523-t001:** Mean diameter, Polydispersity index (PdI), Zeta Potential (ZP) and Entrapment Efficiency (E.E.) of DPH-loaded poly(lactic-co-glycolic) acid (PLGA) nanoparticles (P NPs) and PLGA-Poloxamer nanoparticles (PP NPs). Results are expressed as mean values ± SD (*n* = 3).

Formulation	Mean Diameter ± SD [nm]	PdI ± SD	ZP ± SD [mV]	E.E., [%]
P NP	182 ± 7.1	0.124 ± 0.07	−43.8 ± 2.3	96.8 ± 0.04
PPNP_DPH	120 ± 4.2	0.117 ± 0.02	−39.1 ± 2.0	96.2 ± 0.03

**Table 2 ijms-24-14523-t002:** Glass transition (Tg), peak temperature (Tpeak), onset temperature (Tonset), melting Enthalpy (ΔHm) of unloaded and DPH-loaded poly(lactic-co-glycolic) acid (PLGA) nanoparticles (P NPs) and PLGA-Poloxamer nanoparticles (PP NPs).

NP Formulation	Tg [°C]	T Onset [°C]	T Peak [°C]	ΔHm [J/g]
P	49.4 ± 0.6	-	-	-
PP	-	47.6 ± 0.1	51.6 ± 0.1	33.0 ± 0.6
P (loaded with DPH)	49.3 ± 0.5	-	-	-
PP (loaded with DPH)	-	48.5 ± 3.2	47.8 ± 4.5	20.5 ± 3.4

**Table 3 ijms-24-14523-t003:** Liver results; poly(lactic-co-glycolic) acid (PLGA) nanoparticles (P NPs) and PLGA-Poloxamer nanoparticles (PP NPs); nd = not detected.

Samples	Time [h]	Concentration [ng∙mL^−1^]
P NPs	1	nd
PP NPs	1	13.39 ± 15.24
P NPs	4	nd
PP NPs	4	nd

**Table 4 ijms-24-14523-t004:** Kidney results; poly(lactic-co-glycolic) acid (PLGA) nanoparticles (P NPs) and PLGA-Poloxamer nanoparticles (PP NPs); nd = not detected.

Samples	Time [h]	Concentration [ng∙mL^−1^]
P NPs	1	nd
PP NPs	1	8.86 ± 2.65
P NPs	4	nd
PP NPs	4	<LOD

**Table 5 ijms-24-14523-t005:** Lung results; poly(lactic-co-glycolic) acid (PLGA) nanoparticles (P NPs) and PLGA-Poloxamer nanoparticles (PP NPs); nd = not detected.

Samples	Time [h]	Concentration [ng∙mL^−1^]
P NPs	1	nd
PP NPs	1	10.43 ± 0.74
P NPs	4	nd
PP NPs	4	<LOD

**Table 6 ijms-24-14523-t006:** Serum results; poly(lactic-co-glycolic) acid (PLGA) nanoparticles (P NPs) and PLGA-Poloxamer nanoparticles (PP NPs); nd = not detected.

Samples	Time [h]	Concentration [ng∙mL^−1^]
P NPs	1	232.40 ± 62.60
PP NPs	1	253.73 ± 31.35
P NPs	4	201.91 ± 11.11
PP NPs	4	208.48 ± 22.53

## Data Availability

Not applicable.

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
