# Peer review of "Investigating the Effect of Surface Hydrophilicity on the Destiny of PLGA-Poloxamer Nanoparticles in an In Vivo Animal Model"

_ijms, 2023, doi:10.3390/ijms241914523_

Round 1

Reviewer 1 Report

The current manuscript is an interesting study on the in vivo effects of surface hydrophilicity of PLGA NPs. Many relevant assays were performed, and the methodology is both solid and complete. Nevertheless, some alterations should be made before acceptance for publication:

- The choice of PLGA over other polymers should be better supported in the introduction section, namely with reference to studies that have compared this polymer to others; this should also be done for poloxamers;

- The choice of PLGA type “Equimolar PLGA (Resomer® RG504H, poly(D,L-lactide-co-glycolide) (ester termi-191 nated; lactide:glycolide ratio 50:50; Mw = 38-54 kDa)” should be justified, as well as the choice for Poloxamers 407 and 188, since so many other types exist; why are these better or more relevant for the current study?;

- The designation for the Poloxamers should be corrected, since the authors wrote “Poloxamer F127” and “Poloxamer F68”, and this is a mixture of both brand name and compound name; either write Pluronic F127 and Pluronic F68, or Poloxamer 407 and Poloxamer 188 (or both, separately, but not a fusion);

- An abbreviation list should be added;

- Abbreviations should be written in full in Figure and Table captions;

- Figure 2 quality (resolution) should be improved;

- Possible future administration routes for the produced NPs should be discussed, namely non-invasive routes, and respective therapeutic applicability.

Author Response

We thank the referee for his/her comments and suggestions which certainly helped us to improve the quality of the article. Here follows a point-to-point answers.

Reviewer 1

The current manuscript is an interesting study on the in vivo effects of surface hydrophilicity of PLGA NPs. Many relevant assays were performed, and the methodology is both solid and complete. Nevertheless, some alterations should be made before acceptance for publication:

- The choice of PLGA over other polymers should be better supported in the introduction section, namely with reference to studies that have compared this polymer to others; this should also be done for poloxamers;

As suggested by the referee, a more complete description on the choice of polymers used in this study has been added in the introduction of the paper.

- The choice of PLGA type “Equimolar PLGA (Resomer® RG504H, poly(D,L-lactide-co-glycolide) (ester termi-191 nated; lactide:glycolide ratio 50:50; Mw = 38-54 kDa)” should be justified, as well as the choice for Poloxamers 407 and 188, since so many other types exist; why are these better or more relevant for the current study?;

We agree with the referee and, thus, we have added more details about the choice of polymers used in this study in the introduction section of the paper.

- The designation for the Poloxamers should be corrected, since the authors wrote “Poloxamer F127” and “Poloxamer F68”, and this is a mixture of both brand name and compound name; either write Pluronic F127 and Pluronic F68, or Poloxamer 407 and Poloxamer 188 (or both, separately, but not a fusion);

As suggested by the referee, we have fixed this oversight.

- An abbreviation list should be added;

An abbreviation list has been added as suggested by the referee.

- Abbreviations should be written in full in Figure and Table captions;

This point has been done, following the referee comments.

- Figure 2 quality (resolution) should be improved;

Figure 2 quality was improved.

- Possible future administration routes for the produced NPs should be discussed, namely non-invasive routes, and respective therapeutic applicability.

We agree with the referee and accordingly, a bit discussion about NP route od administration has been added to the paper at the end of the Conclusions.

Reviewer 2 Report

1.     The author should perform some statistical analysis on Figure 3.

2.     What about longer release profile of P NPs and PP NPs? 4 h seems too short release time.

3.     What is the toxicity of the P NPs and PP NPs both in vitro and in vivo? The author should provide at least one cytotoxicity result on 1 cell line, and in vivo weight change to show the toxicity of the NPs.

4.     The author stated that PP NPs could be only detected inside the blood vessels. What about longer period of time? the author only did 1 and 4 h injection, that is too short period of time.

English is fine to understand. 

Author Response

We thank the referee for his/her comments and suggestions which certainly helped us to improve the quality of the article. Here follows a point-to-point answers.

Reviewer 2

The author should perform some statistical analysis on Figure 3.

Statistical analysis on Figure 3 was added as suggested by the referee.

 What about longer release profile of P NPs and PP NPs? 4 h seems too short release time.

We thank the referee for this observation. Actually, in such a small animal model, 4 hours represents the average circulation time of nanometric particles and, generally, the biodistribution in the tissues begins already after 30 minutes. Indeed, there are many examples in the literature of similar studies in which the biodistribution of nanoparticles in mice was studied, in brackets a couple of examples (Journal of Controlled Release, Volume 359, July 2023, Pages 257-267; Acta Biomaterialia, Volume 7, Issue 12, December 2011, Pages 4169-4176).

What is the toxicity of the P NPs and PP NPs both in vitro and in vivo? The author should provide at least one cytotoxicity result on 1 cell line, and in vivo weight change to show the toxicity of the NPs.

      In two recent publications, we have shown that NPs were not toxic and curcumin encapsulated in such NPs was released for days and this allowed a block of mesothelioma cancer cells amplification in the G0/G1 phase of the cell cycle for up to 72 hours [10] [11]. As suggested by the referee, we highlighted this aspect in the introduction of the paper.

The author stated that PP NPs could be only detected inside the blood vessels. What about longer period of time? the author only did 1 and 4 h injection, that is too short period of time.

We thank the referee for this observation; however, the aim of the present work was to evaluate whether the different surface properties could influence the biodistribution in healthy animal model. Future studies, still ongoing, are aimed at evaluating the different biodistribution of NPs, loaded with an appropriate drug, in pathological animal models in order also to evaluate their therapeutic potential in the long term.

Round 2

Reviewer 1 Report

The manuscript is suitable for publication.

Author Response

Thank you,

regards. 

Reviewer 2 Report

All comments are addressed. 

Author Response

Thank you,

regards.